# Determination of Geographical Origin of Southern Shaanxi Congou Black Teas Using Sensory Analysis Combined with Gas Chromatography–Ion Mobility Spectrometry

**DOI:** 10.3390/foods13233904

**Published:** 2024-12-03

**Authors:** Fei Yan, Xiaohua Chen, Dong Qu, Wei Huang, Lijuan He, Tian Wan, Lijun Zhang, Qi Wang, Ching Yuan Hu

**Affiliations:** 1Shaanxi Provincial Bioresources Key Laboratory, Shaanxi University of Technology, Hanzhong 723000, China; yanfei@snut.edu.cn (F.Y.); qudong@snut.edu.cn (D.Q.); wtian@snut.edu.cn (T.W.); 2Qinba State Key Laboratory of Biological Resources and Ecological Environment, Qinling-Bashan Mountains Bioresources Comprehensive Development C.I.C. College of Biological Science and Engineering, Hanzhong 723001, China; 3Hanzhong Food and Drug Inspection and Testing Center, Hanzhong 723000, China; huangwei9950@163.com (W.H.); taozijiahome@163.com (L.H.); 4Ankang R&D Center for Se-Enriched Products, Ankang 725000, China; 13379410321@163.com (L.Z.); 18946612118@163.com (Q.W.); 5Department of Human Nutrition, Food and Animal Sciences, College of Tropical Agriculture and Human Resources, University of Hawaii at Manoa, 1955 East-West Road, AgSci. 415J, Honolulu, HI 96822, USA

**Keywords:** Shaanxi, congou black tea, aroma type, sensory evaluation, GC-IMS, geographical origin

## Abstract

Southern Shaanxi is one of China’s high-quality congou black tea production areas. However, the differences in geography, cultivation, and management techniques and production processes lead to uneven qualities of southern Shaanxi congou black tea in different production areas. This study used sensory analysis combined with gas chromatography–ion mobility spectrometry (GC-IMS) to determine southern Shaanxi congou black teas’ geographical origin and volatile fingerprints to prevent economic losses caused by fraudulent labeling. A total of 61 volatile compounds were identified and quantified by GC-IMS. Three main aroma types were found by sensory analysis coupled with significant difference analysis, and a clear correlation between volatile compounds, aroma type, and geographical origin was found by sensory and gallery plot analysis. The black tea with a green/grassy-roast aroma type was mainly distributed in production areas with an altitude of 400–800 m and 1-pentanol, cyclohexanone, 1-penten-3-one, 2-heptanone, dihydroactinidiolide and butyrolactone were the key aroma markers. The black teas produced in production areas with an altitude of 800–1000 m mainly presented strong honey and caramel-like aromas, and sotolone, furaneol, and phenylacetaldehyde played an important role. These results will be helpful for discriminating black tea from different tea production areas in southern Shaanxi.

## 1. Introduction

The black teas obtained from the processed shoots of Camellia sinensis are the most consumed tea beverage globally for their pleasant flavor, and China is one of the leading producers [1]. Based on the different manufacturing methods, Chinese black teas are divided into three categories: souchong black teas, congou black teas, and broken black teas [2]. Among them, congou black teas are traditional export commodities exclusively produced in China.

Southern Shaanxi, located in the famous Qinling–Daba Mountains, is the geographical and climatic division belt that separates northern and southern China. Southern Shaanxi is also the northernmost tea-growing region in China (Figure 1). The tea plants in the areas are usually at an elevation of 500–1000 m and latitude of 32–37° with 800–1300 mm of annual rainfall, 2000 h of annual sunshine, and an annual average temperature of about 14 °C [1]. China’s unique geographical and climatic conditions are considered one of the most suitable areas for producing congou black tea.

As is well known, the altitude, cultivation and management techniques, production processes, and storage conditions of tea plant growing areas have important impacts on the tea quality [3]. These differences lead to uneven levels of congou black tea quality from different production areas in southern Shaanxi. However, these black teas are often collectively referred to as southern Shaanxi black tea. Therefore, to avoid economic losses caused by fraudulent labeling and provide quality assurance for buyers and sellers, it is vital to identify the geographical origin of southern Shaanxi congou black teas.

Aroma is the most crucial factor affecting tea quality, market value, and consumer acceptance [4]. In recent years, the unique sensory characteristics of black tea from different geographical regions have attracted more attention due to their important role in origin identification, quality control, avoiding label fraud, and improving economic value [5,6,7]. Sensory evaluation is a commonly used method for tea aroma identification. However, the traditional evaluation results based on skilled tasters were usually subjective and lacked objective data support [8]. Recently, several analytical techniques have been used for the geographical tracing of tea. These techniques include the electronic tongue, nuclear magnetic resonance spectroscopy, near-infrared spectroscopy, gas chromatography–mass spectrometry, and gas chromatography–ion mobility spectrometry (GC-IMS) [7,9,10]. Compared to the traditional GC-MS, GC-IMS is a new gas-phase separation and detection technology that combines the high separation capacity of GC and the fast response of ion mobility spectrometry (IMS). GC-IMS has the advantages of low detection limit, short analysis time, and no need for concentration and enrichment of samples. It has been widely used in origin identification, wine age analysis, quality control, aroma component identification, and flavor fingerprinting [8,11,12].

In this study, the aroma type and volatile components of congou black tea from different production areas in southern Shaanxi were identified by sensory analysis and GC-IMS, respectively. Furthermore, the relationship between aroma type and geographical origin was analyzed by cluster analysis. Last, their flavor fingerprints were established. This study aims to quickly distinguish southern Shaanxi black tea from different production areas using sensory analysis combined with gas chromatography–ion mobility spectrometry to avoid labeling fraud and provide guidance for producing black tea with characteristic flavors under the most suitable environmental conditions.

## 2. Materials and Methods

### 2.1. Black Tea Samples

A total of 14 representative southern Shaanxi congou black tea samples were collected from seven main production areas, including Zhenba County, Mianxian County, Nanzheng County, Lueyang County, Ningqiang County, Xixiang County, and Chenggu County. Among them, Xixiang County is the largest tea-producing area, followed by Mianxian County, Zhenba County, and Nanzheng County, while Lueyang, Chenggu, and Ningqiang Counties are the smallest (Figure 1). The geographical origins of black tea samples were identified by the Tea Product Quality Supervision and Inspection Center of Shaanxi Province (Table 1). All tea leaves were picked in the first week of June 2023 and manufactured following the unified processing guidelines (Technical regulation for processing of black tea, GB/T 35810-2018 [13], processing steps: fresh leaves, withering, rolling, oxidizing, drying, primary tea, screening, drying, black tea).

### 2.2. Chemicals

The reference compounds 2-methyl butanal (98%), 1-penten-3-one (99.5%), pentanal (97%), propyl acetate (99.5%), 3-hydroxybutan-2-one (95%), butyrolactone (99.5%), methyl isobutyl ketone (98%), β-damascenone (97%), (E)-2-pentenal (99.5%), 1-pentanol (98%), dihydroactinidiolide (95%), hexanal (97%), butyl acetate (95%), methyl hexadecanoate (98%), cis-jasmone (97%), furaneol (98%), tetrahydrofuran (99.5%), 3-methyl butanal (97%), 2-acetylpyrrole (95%), 2-methylpyrazine (99%), 2-butoxyethanol (95%), heptanal (96%), butanolide (97%), β-ionone (97%), (E)-2-heptenal (98%), (E)-2-hexenal (96%), benzaldehyde (96%), 5-methylfurfural (95%), and 3-octanol (95%) were purchased from ANPEL Laboratory Technologies Inc. (Shanghai, China). 6-methyl-5-hepten-2-one (98%), 2-furanmethanethiol (96%), 2-pentylfurane (97%), (E, E)-2,4-heptadienal (97%), benzene acetaldehyde (97%), (Z)-3-hexenyl benzoate (96%), butanoic acid (97%), 2-methyl-2-pentenal (96%), furfural (96%), indole (98%), methyl jasmonate (95%), coumarin (99%), 2-furanmethanol (96%), 2-methyl-3-furanthiol (98%), 2-acetylpyrazine (96%), 2-heptanone (98%), cyclohexanone (96%), 2-methylbutanoic acid ethyl ester (96%), methyl acetate (95%), 1-propanol (97%), 2-methyl-2-propenal (95%), 2-butanone (97%), butanal (96%), ethyl acetate (95%), linalool oxide (94%), dihydromyrcenol (96%), linalool (95%), sotolone (98%), (E,E)-2,4-octadienal (96%), nonanal (98%), and methyl salicylate (95%) were purchased from Beijing Bellwether Technology Co., Ltd. (Beijing, China).

### 2.3. GC-IMS Analysis

The volatile compounds in the black tea samples were analyzed using a GC-IMS (Flavorspec, G.A.S. Instrument, Dortmund, Germany) with an MXT-5 column (15 m × 0.53 mm × 1 μm) (Restek, PA, USA). The operating conditions of GC-IMS were conducted following a published method [9] with minor modifications. First, 3.0 g tea samples were brewed in 200 mL of 95 °C distilled water for 5 min, and then 10 mL of tea infusion was poured into a 30-mL headspace vial with a magnetic screw seal cover. Then, the samples were incubated at 80 °C for 15 min. 500 μL of the headspace gas was injected with an automatic headspace sampling unit into the injector at 85 °C using splitless injection. The column was 60 °C, IMS temperature was set to 45 °C. The drift gas flow was set to a constant 150 mL/min flow rate. Nitrogen (99.999% purity) was used as a carrier gas at a 2 mL/min flow rate. Then, the rate was increased to 2 mL/min for 2 min and 100 mL/min for 20 min, and the flow was stopped. All tests were repeated three times, and the retention indices (RI) were calculated using n-ketones C4–C9 (Sinopharm Chemical Reagent Beijing Co., Ltd., Beijing, China). The volatile compounds were identified by comparing RI and the drift time (Dt; the time required for ions to reach the collector through the drift tube) of the reference compound.

### 2.4. Aroma Characteristics Analysis

A quantitative descriptive analysis (QDA) method [2] was applied to identify the aroma type of the southern Shaanxi black teas, using a 6-point hedonic scale (e.g., 0 = weak/slightly to 5 = strong/very). First, the QDA was carried out using the methods described in GB/T23776-2009 [14] (Methodology of sensory evaluation of tea) and Chen et al. [15,16] in a sensory panel room at 21 ± 1 °C. Second, seven panelists (four females and three males, aged 20 to 23 years) were employed for the sensory evaluation. Before the formal experiment began, they underwent a 6-week sensory training program. The training of the sensory panel was conducted in five sessions. The key descriptors selected for sensory evaluation included roasted, caramel-like, honey-like, green/grassy, and floral, as shown in Table 2. At first, the aroma attributes of the 14 black tea infusions were identified by the panel members. Then, the panel members intensively discussed and selected aroma attributes according to the Standardization Administration of the People’s Republic of China (SAC) and General Administration of Quality Supervision, Inspection, and Quarantine of the People’s Republic of China (AQSIQ) GB/T 14487-2008 [17] and Chen et al. [15,16]. Third, the intensity of each selected representative aroma attribute in 14 black tea infusions was evaluated using a 6-point hedonic scale (e.g., 0 = weak/slightly to 5 = strong/very). Finally, the aroma attributes with significant differences (*p* < 0.05) were identified as the aroma type of black tea. Each test was conducted three times, and the final sensory score of each aroma attribute was the average of the scores from the panelists, which was used to plot in a radar diagram. The experimental procedures used in this study were approved by the Management Committee of Shaanxi Provincial Key Laboratory of Bioresources (Approval No. 2023-11).

### 2.5. Statistical Analysis

Each experiment was replicated three times. Data are expressed as mean with standard deviation. The hierarchical cluster and differences in intensity between aroma attributes were analyzed using SPSS v21.0 (SPSS Inc., Chicago, IL, USA). The gallery plot was drawn using the gallery plot plug-in components of GC-IMS.

## 3. Results and Discussion

### 3.1. Aroma Types of Southern Shaanxi Congou Black Teas

As shown in Figure 2, a total of seven different sensory attributes, including roasted, caramel-like, green/grassy, floral, fruity, honey-like, and woody aromas, were identified in southern Shaanxi congou black teas by QDA analysis. The significant difference analysis shows that ZB039 and NZ066 black tea samples had the most robust “caramel-like” aroma with a sensory score of 3.8–5.0 and were identified as caramel-like type. In contrast, the MX013 black tea samples possessed the most potent “honey-like” aroma, with a sensory score of 5.0, and were identified as the honey-like type (Figure 2a). On the other hand, the XX002, 007, and 038, LY016, ZB024, NZ027, CG012 and 048, NQ009, MX004 and LY047 black tea samples exhibited more potent “green/grassy” and/or “roasted” aromas and were identified as the green/grassy–roast aroma type (Figure 2b,c).

Compared with taste and texture, aroma can distinguish important quality characteristics of tea from different geographical regions [3]. For example, Darjeeling and Assam black teas from India have musky and smoky aroma characteristics. Ceylon black teas from Sri Lanka have a characteristically floral and sweet odor [6,18]. In China, Keemun congou black tea from Anhui Province is well known for its floral and honey aroma. Tanyang congou black tea from Fujian Province has a sweet, potent odor, and Yixing congou black tea from Jiangsu Province shows a robust roasted and malty aroma. In contrast, the aroma type of Dianhong congou black tea from Yunnan Province is known to have a green/grassy aroma [2]. Our results show that southern Shaanxi black teas exhibited unique aroma characteristics and rich aroma diversity.

### 3.2. Volatile Profile Analysis of Southern Shaanxi Congou Black Teas by GC-IMS

GC-IMS was used to identify the volatile compounds of southern Shaanxi congou black teas. The visual two-dimensional (2D) spectrograms of volatile compounds are shown in Figure 3. They show that most signals appeared within a hold time of 100–300 s and a drift time of 8–12 ms. The signal strength of the volatile compounds was presented by color in the 2D spectrogram, and the darker red color represents the higher quantity of the compound. In contrast, the lighter red color means that the quantity was lower. The graphical results show that the peak intensities differed significantly among the black teas, indicating that the amount of these compounds differed in different black teas. GC-MS uses retention time and ion fragment characteristics to identify and characterize compounds. In contrast, GC-IMS uses retention time and drift time to accomplish the same task. Thus, GC-IMS is a much easier way to separate and identify isobaric and isomeric acids, compounds with similar retention times [19].

As shown in Figure 4, a total of 61 volatile compounds were identified by GC-IMS, including 17 aldehydes (44.1%), 11 ketones (12.9%), six alcohols (2.8%), ten esters (24.0%), 15 heterocycles (16.1%), one ether (0.1%), and one acid (0.1%). Most of them produced multiple signals or spots (dimers or trimers), mainly attributed to the higher proton affinity of the compounds than water during ionization [20,21] (Appendix A).

The total alcohol contents identified in ZB039 and LY047 black tea samples (10.8% and 9.6%, respectively) were higher (*p* < 0.05) than those found in other black tea samples (0.4~8.9%). Moreover, the ZB039 black tea sample also contained high acids (9.9%). The total heterocycle intensity values in MX004, MX013, and LY047 black tea samples (14.3%, 12.3, and 11.4%, respectively) were higher (*p* < 0.05) than in other black tea samples (0.7~8.2%). In addition, a higher acid intensity value was also found in MX004 and LY047 black tea samples. The total intensity values of ketones and esters were not different in all southern Shaanxi black teas except the LY016 black tea sample (*p* < 0.05), which had the lowest content.

Most of the signals of volatile compounds in Chinese black teas were distributed within a retention time ranging from 100 to 500 s and a drift time of 1.0–1.5 ms [12,22]. Similar results were obtained by 2D plots of southern Shaanxi congou black teas, indicating their similar aroma components. For example, linalool, nonanal, (E,E)-2,4-heptadienal, benzaldehyde, octanal, heptanal, and hexanal, identified in southern Shaanxi black teas, were also found in black teas from other regions of China [22]. However, the main components between them were significantly different. The main compounds in southern Shaanxi black teas are dominated by (E)-2-hexenal (ZB039 sample), hexanal (ZB039 sample), 2-methylbutanal (MX004, NZ066 and NQ009 samples), 3-methylbutanal (all samples), 2-butanone (all samples), propyl acetate (XX002, XX007, XX038, ZB024, CG012, CG048, NZ027, NZ066), and 2-methylbutanoic acid ethyl ester (all samples). In contrast, black teas from other regions of China are composed of 1-propenyl propyl disulfide, linalool, (E)-2-hexen-1-ol, and acetone [12,22]. This difference might be attributed to different geographical environments, variety, cultivation techniques, picking seasons, manufacturing processes, and storage conditions [3].

GC-IMS is considered the most convenient method for food flavor analysis due to its low vacuum requirements, simple sample pretreatment, and short analysis time [19]. In addition, compared with the popular GC-MS, GC-IMS also provides additional volatile compound information for understanding food flavor. For example, a large number of compounds, such as (E,E)-2,4-heptadienal, octanal, (E,E)-2,4-octadienal, pentanal, 1-penten-3-one, pentanol, and propyl acetate, were identified in southern Shaanxi black teas by GC-IMS, which were not identified in our previous GC-MS analyses [15,16].

### 3.3. Relationship Between Aroma Type, Volatile Components and Geographical Origin

As shown in Figure 5, a good relationship was identified between aroma type and volatile compounds. Cluster analysis based on volatile contents divided the black teas with similar aroma types into the same groups. In addition, a clear correlation between aroma type and geographical origin was found. The black teas with the green/grassy–roast aroma type were mainly distributed in Lueyang, Xixiang, and Chenggu Counties with an altitude of 400–800 m, while the black teas produced in Mianxian and Zhenba Counties, with an altitude of 800–1000 m, mainly presented strong caramel and honey-like aromas. Moreover, the black teas with green/grassy and roast aroma types were further grouped into subcluster 1 and subcluster A, and the black teas with caramel-like aroma types were grouped into subcluster 2 and cluster III, which indicates that there were significant differences in volatile components between black teas with the similar aroma type.

To visually display the differences between the compounds in the three aroma types of black teas, gallery plot processing using the VOCal software 04.07 was performed. The results show apparent differences among the three aroma types of black teas (Figure 4). Specifically, 1-pentanol, cyclohexanone, 1-penten-3-one, 2-heptanone, dihydroactinidiolide, and butyrolactone in black tea with green/grassy–roast aroma type (subcluster 1) were higher (*p* < 0.05) than those in other aroma type black teas. For the black teas in subcluster A, 2-methyl-2-propenal, furfural, butanal, 3-methylbutanal, 2-methylbutanal, 5-methylfurfural, 2-furanmethanol, 1-propanol, 2-acetylpyrazine, coumarin, methyl hexadecanoate, hexanal, (Z)-3-hexenyl benzoate, and methyl acetate present a significantly higher content (*p* < 0.05) than other southern Shaanxi black teas. The aroma compounds are the most important factor influencing tea’s aroma characteristics, and they vary by elevation [22]. For example, the content of compounds with a green/grassy odor note, including 1-penten-3-ol, Z-3-hexenal, E-2-hexenal, Z-2-pentenol, and pentanol, was higher in the black tea from low elevations. While the aroma compounds with the floral and fruity aroma, such as linalool and its oxide, β-cyclocitrol, geraniol, nerolidol and benzyl alcohol, were present at a higher level in high-elevation black teas [2,23,24,25]. Similarly, the low-elevation southern Shaanxi black teas contained numerous volatile compounds with green/grassy odor notes (400–800 m above sea level). These included (Z)-3-hexenyl benzoate, (E,E)-2,4-heptadienal, (E)-2-heptenal, heptanal, (E)-2-hexenal, hexanal, and (E)-2-pentenal. The aroma compounds associated with a green/grassy odor were mainly produced by biological and abiotic oxidation of unsaturated fatty acids in tea leaves [26]. Therefore, the higher temperature and oxygen content in low-elevation areas might be more conducive to synthesizing these compounds. Although the content of these compounds was low, they can readily be perceived due to their high volatility, synergistic effect, and additive action between them [13,27], which might make the aroma characteristics of black tea from low elevation more easily dominated by the green/grassy aroma. Moreover, due to the significantly lower content of aroma compounds in tea leaves grown at low altitudes than at high altitudes, increasing the roasting temperature and time of tea processing to enhance black tea aromas is a method commonly used by local tea manufacturers, which may lead to more roasted aromas being produced through the Maillard reaction in black tea grown in southern Shaanxi at low-altitude areas (400–800 m) [26].

Caramel and honey-like aromas were the characteristic sensory attributes of higher-elevation black teas, such as Darjeeling and Keemun [6,18]. The sensory attributes were also found in southern Shaanxi black tea from Mianxian and Zhenba Counties at an altitude of 800–1000 m. Gallery plot analysis exhibited higher contents (*p* < 0.05) of sotolone (caramel-like odor note) and phenylacetaldehyde (honey-like odor note) in black teas (subcluster III). Moreover, the strong caramel aroma was also found in the black tea from Nanzheng County (subcluster 2), and the content (*p* < 0.05) was dominated by furaneol (caramel-like odor note). These compounds were identified as the key contributors to caramel-like and honey-like aromas of southern Shaanxi black tea in our previous research [15,16]. In addition, higher contents of floral aroma compounds such as linalool and its oxides have also been observed in high-elevation black tea from Zhenba County. The synthesis and accumulation of aroma compounds with honey and floral aromas were highly influenced by altitude, and the higher altitude, the higher levels of these compounds, which was mainly due to the transcription levels of related structural genes [28]. Compared to our previous studies [15,16], this work provides aroma markers for southern Shaanxi congou black teas from different geographical origins and comprehensive information on quality assessment. Moreover, southern Shaanxi congou black tea rich in furaneol (caramel-like odor note) was first identified using GC-IMS technology in this study, which will provide richer fingerprint information for the geographical origin identification of black teas from southern Shaanxi.

## 4. Conclusions

In this study, we used sensory analysis combined with gas chromatography–ion mobility spectrometry to distinguish the southern Shaanxi congou black tea made from plants grown at different geographical origins. A total of 61 common aroma compounds were identified and quantified by GC-IMS. The sensory analysis and significant difference analysis confirmed differences in the aroma character of southern Shaanxi congou black tea from different geographical origins, and three main aroma types were found, including caramel-like, honey-like, and green/grassy–roast aromas. Among them, the black teas with the green/grassy–roast aroma type were mainly distributed in Lueyang, Xixiang, and Chenggu Counties at an altitude of 400–800 m, while the black teas produced in Mianxian and Zhenba Counties at an altitude of 800–1000 m mainly presented strong honey and caramel-like aromas. The concentration difference of volatile compounds via the gallery plot analysis identifies the aroma markers of aroma types and distinguished geographical origin. Based on data analysis, 1-pentanol, cyclohexanone, 1-penten-3-one, 2-heptanone, dihydroactinidiolide, and butyrolactone were the key aroma markers in southern Shaanxi congou black tea with the green/grassy–roast aroma. Sotolone, furaneol, and phenylacetaldehyde played an important role in black teas with caramel- and honey-like aromas, respectively. Our results will be helpful in discriminating black tea from different tea production areas in southern Shaanxi. Our method can also be used for teas from other tea-producing regions.

## Figures and Tables

**Figure 1 foods-13-03904-f001:**
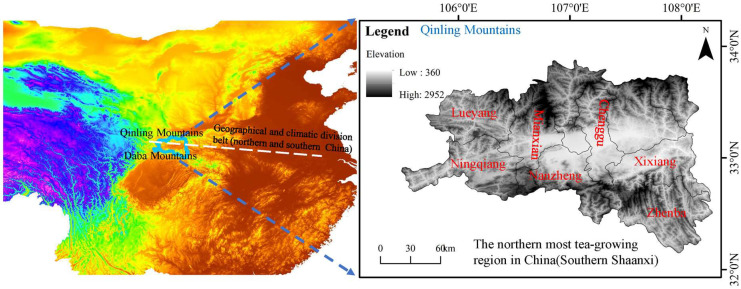
Geographical growing areas of southern Shaanxi congou black tea.

**Figure 2 foods-13-03904-f002:**
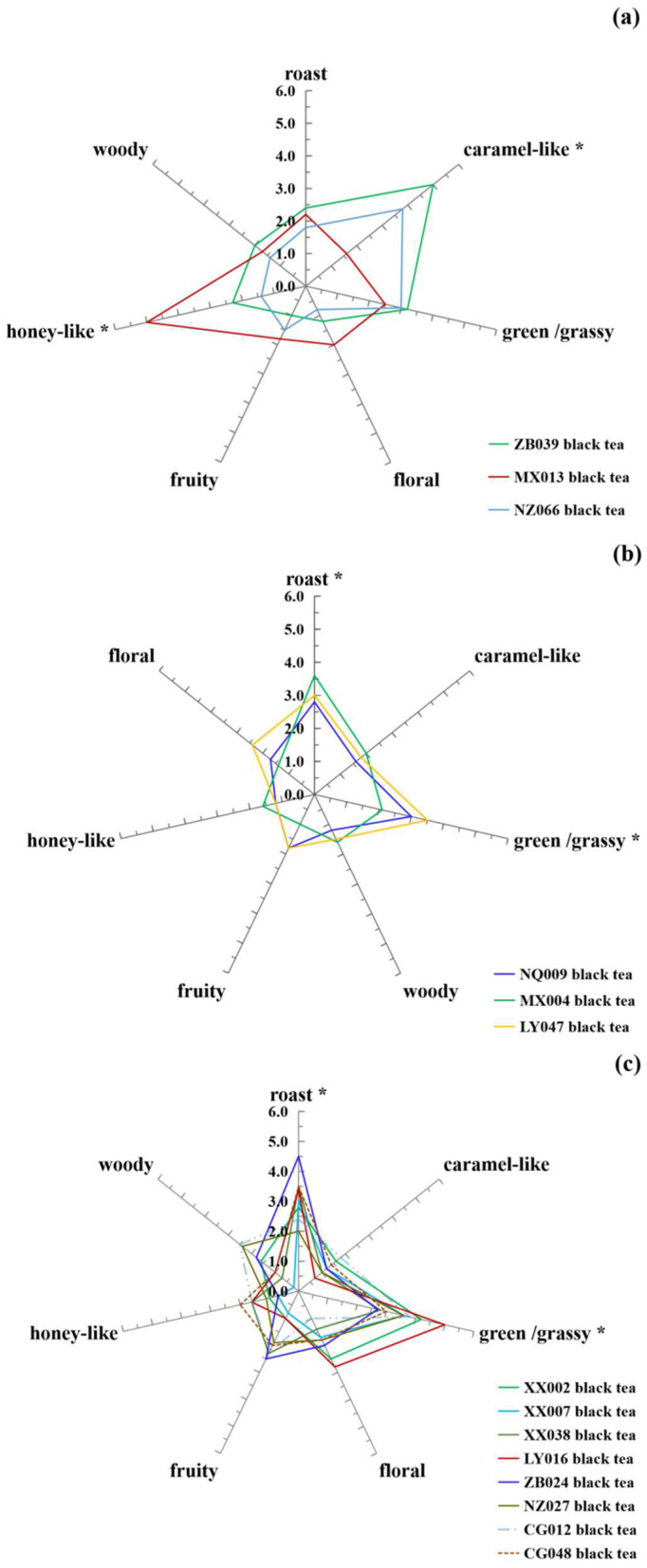
Diagram of odor perceptions and intensities of southern Shaanxi congou black tea from different tea-producing areas obtained from sensory analysis. (**a**) Sensory analysis diagram of ZB039, MX013 and NZ066 balck tea. (**b**) Sensory analysis diagram of NQ009, MX004 and LY047 balck tea. (**c**) Sensory analysis diagram of XX002, XX007, XX038, LY016, ZB024, NZ027, CG012 and CG048 balck tea. * Significant difference at *p* < 0.05.

**Figure 3 foods-13-03904-f003:**
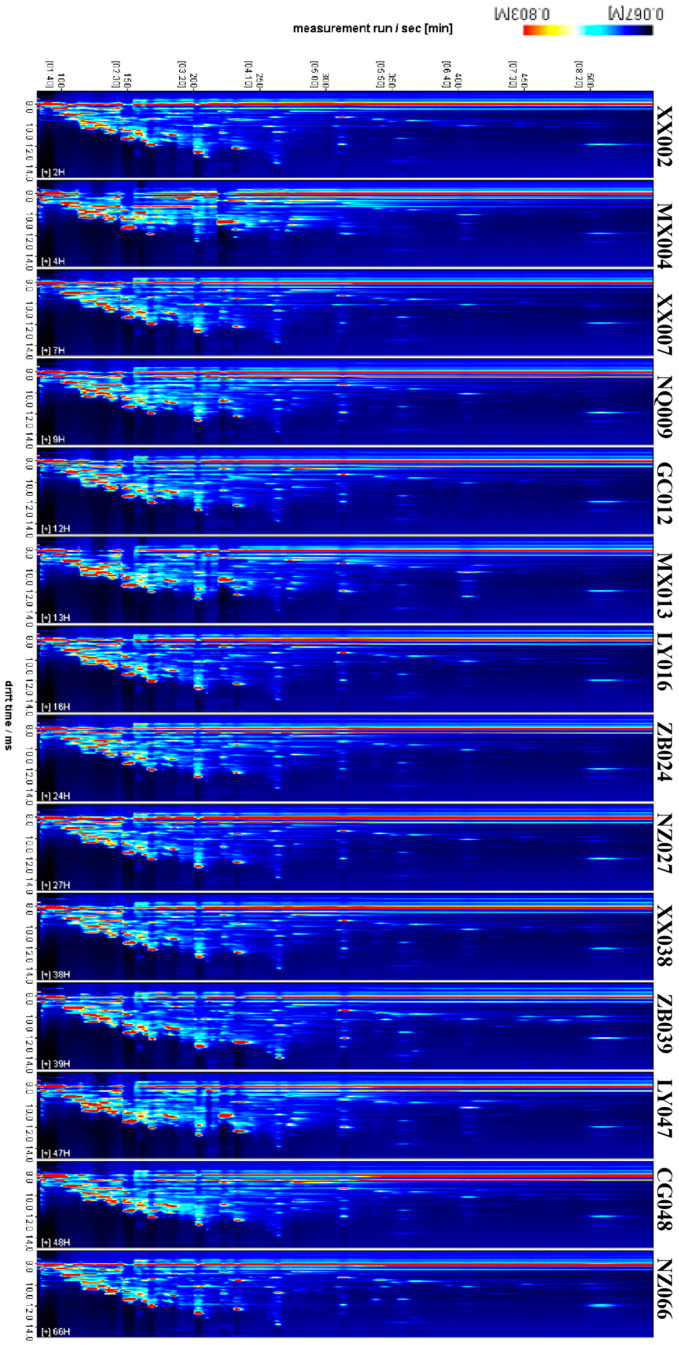
Two-dimensional spectra of volatile compounds of congou black tea from different tea-producing areas in southern Shaanxi.

**Figure 4 foods-13-03904-f004:**
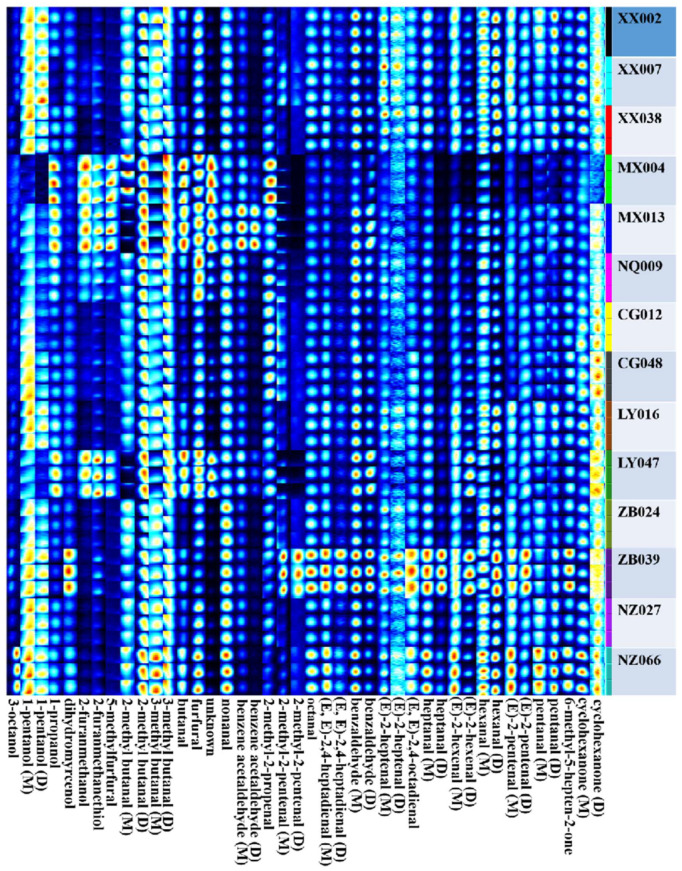
Gallery plot of the volatile compounds of congou black tea from different tea-producing areas in southern Shaanxi.

**Figure 5 foods-13-03904-f005:**
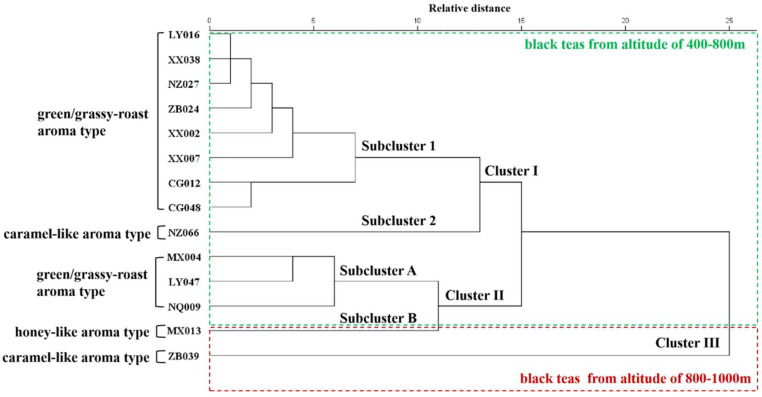
The dendrogram was obtained by cluster analysis of the volatile components in southern Shaanxi congou black teas using the systematic clustering method.

**Table 1 foods-13-03904-t001:** Information of representative congou black tea samples from southern Shaanxi.

Production Area	Tea Sample	Annual Rainfall (mm)	Annual Sunshine (h)	Annual Average Temperature (°C)	Altitude (m)	Latitude
Zhenba County	ZB039	1230	1183	15	1100	32°05′~32°08′
	ZB024	1100	1651	13	685	
Mianxian County	MX013	850	1670	13.9	1040	32°53′~33°38′
	MX004	840	1801	14.2	750	
Nanzheng County	NZ066	925	1621	14.3	620	32°24′~33°07′
	NZ027	934	1615	14.8	420	
Lueyang County	LY004	777	1558	13.6	730	33°16′~33°27′
	LY016	775	1537	13.2	650	
Ningqiang County	NQ009	1812	1589	12.9	687	32°37′~33°12′
Xixiang County	XX038	923	1698	14.4	600	32°32′~33°14′
	XX007	910	1685	14.2	650	
	XX002	910	1690	14.4	452	
Chenggu County	CG048	843	1518	14.3	540	32°45′~33°40′
	CG012	840	1520	14.4	520	

**Table 2 foods-13-03904-t002:** The definitions of sensory attributes for black teas.

Sensory Attribute	Definition	Reference Standard
Roasted	Aroma associated with burning, toasting, roasting, wood fire, and smoke	10 mg/L 2-acetylpyrazine aqueous solution
Caramel-like	Sugar after high temperatures	10 mg/L 4,5-Dimethyl-3-hydroxy-2,5-dihydrofuran-2-one aqueous solution
Honey-like	Pleasant honey-like note	10 mg/L phenylacetaldehyde aqueous solution
Green/grassy	Disagreeable and inharmonious scent of green grass, crushed leaves, and unripe fruit	10 mg/L (Z)-3-hexen-1-ol aqueous solution
Floral	Perfume of fresh flowers	0.5 mg/L (R)-linalool aqueous solution

## Data Availability

The original contributions presented in the study are included in the article/Appendix A; further inquiries can be directed to the corresponding authors.

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
