# Peer review of "Determination of Geographical Origin of Southern Shaanxi Congou Black Teas Using Sensory Analysis Combined with Gas Chromatography–Ion Mobility Spectrometry"

_foods, 2024, doi:10.3390/foods13233904_

Round 1

Reviewer 1 Report

Comments and Suggestions for Authors

The authors studied the sensory analysis and GC-IMS identification to determine the geographical origin of Southern Shaanxi Congou Black Teas. The topic is interesting, the methodology is well-designed, and the manuscript is well-written. The research work is novel, however, some revisions need to be addressed before the manuscript can be considered for Foods

  1. Page 2, Introduction, author should mention why GC-IMS is better than traditional GC-MS or GC-O-MS.
  2. On page 3, line 91, the authors should mention the steps involved in the processing of black tea varieties.
  3. Lines 121-122: "The column temperature was 60°C, IMS temperature was set to 45°C"... Is this an isothermal temperature, or did the authors use a temperature ramp program?
  4. On page 4, "Aroma Characteristics Analysis," the authors should mention the key descriptors selected for sensory evaluation and which reference or standard compounds were used to train their odor descriptions.
  5. Page 4, 2.4. The author should mention the time (hours or sessions) spent on the training of sensory panel.
  6. On page 4, lines 141-142, the authors mention using a 6-point hedonic scale, however, in the following line, the description refers to a scale from 0 to 5 points. This should be clarified.
  7. On page 4, in section "3.1. Aroma Types of Southern Shaanxi Congou Black Teas," the author should mention the sensory scores for the most prominent aroma descriptors instead of only writing the intensities.
  8. On page 5, line 173, "As our results show..." please check for grammatical errors.
  9. On page 6, lines 183-184, the authors first state, "The peak signal distribution was similar for all samples, which indicates that the volatile compounds in black teas from different aroma types were the same". However, in lines 189-191, they write, "The graphical results show that the peak intensities differed significantly among the black teas, indicating that the amount of these compounds differed in different black teas". The authors should clarify this apparent contradiction.
  10. On page 9, lines 223-224. However, the main components between them were significantly different….”. The authors should clarify which samples of Southern Shaanxi black teas they are referring to here.
  11. On page 12, line 316: "The concentration difference of volatile compounds..." Which figure shows the concentration differences among volatile compounds? Did the authors calculate the concentrations of the 61 volatile compounds?

Reviewer 2 Report

Comments and Suggestions for Authors

The authors in the manuscript FOODS-3305027 analyzed the volatile compounds in the congou black tea extracts and investigated associations with geographical region of cultivation. A total of 61 volatile compounds were identified using the GC-IMS method and the teas were divided into several groups according to the aroma profile. Finally, the relationship between aroma type, volatile components, and geographic origin was examined.

The topic of the manuscript is consistent with the scope of the journal. In general, this paper may be of value for readers, but I have a few suggestions for improving the manuscript that are listed below. The manuscript should be revised (minor revisions).

1)   Figure 3 is redundant, as the same information is presented more clearly in Figure 4.

2)   Line 184 and Figure 4: The drift time given in the text does not correspond to Figure 4. Axes are not easy to read, but the most peaks is in drift time 8-12 ms (in the text 1-1.5 ms).

3)   Chapter 3.2.: The percentages provided for each group of substances are not explained. Is it percentage representation or concentration representation? On the line 208 is indicated “…content were higher…”. Which method was used to determine the content of analytes? The measured values should be provided, at minimum, in the supplementary materials.
